# Sparse Fréchet sufficient dimension reduction via nonconvex optimization

Jiaying Weng[1],[*] Chenlu Ke[2], Pei Wang[3]

[1]Bentley University, [2]Virginia Commonwealth University, [3]Miami University

jweng@bentley.edu, kec2@vcu.edu, wangp33@miamioh.edu

In the evolving landscape of statistical learning, exploiting low-dimensional structures, particularly for non-Euclidean objects, is an essential and ubiquitous task with wide applications ranging from image analysis to biomedical research. Among the momentous developments in the non-Euclidean domain, Fréchet regression extends beyond Riemannian manifolds to study complex random response objects in a metric space with Euclidean features. Our work focuses on sparse Fréchet dimension reduction where the number of features far exceeds the sample size. The goal is to achieve parsimonious models by identifying a low-dimensional and sparse representation of features through sufficient dimension reduction. To this end, we construct a multitask regression model with synthetic responses and achieve sparse estimation by leveraging the minimax concave penalty. Our approach not only sidesteps inverting a large covariance matrix but also mitigates estimation bias in feature selection. To tackle the nonconvex optimization challenge, we develop a double approximation shrinkage-thresholding algorithm that combines a linear approximation to the penalty term and a quadratic approximation to the loss function. The proposed algorithm is efficient as each iteration has a clear and explicit solution. Experimental results for both simulated and real-world data demonstrate the superior performance of the proposed method compared to existing alternatives.

## 1. Introduction

Non-Euclidean objects, such as images, shapes, tensors, and random densities, have captivated the attention of data scientists and statisticians as high-throughput data collection and generation continue to expand. Fréchet regression model, introduced by [1], analyzes a random object response in a metric space and a predictor vector in a Euclidean space. Advances to handle high-dimensional predictors in Fréchet regression include the variable selection method proposed by [2] and the single-index model introduced by [3]. In this paper, we consider a multi-index model assuming that a few linear combinations of the predictors contain all the relevant information about the non-Euclidean outcome. Let $X \in \mathbb{R}^p$ be the predictor vector, and let $Y$ be a random object in a metric space. We assume that the joint distribution $F(X, Y)$ and the conditional distribution $F(Y|X)$ exist. The multi-index model can be stated as: $Y = f(\beta_1^\top X, \beta_2^\top X, \ldots, \beta_d^\top X, \epsilon)$, where each $\beta_i \in \mathbb{R}^{p \times 1}$ denotes a projection vector, $d$ is the structural dimension assumed to be significantly smaller than $p$, $\epsilon$ denotes the random error, and $f$ is an unknown link function that maps from Euclidean space to a metric space. Our objective is to find the $d$-dimensional representation $(\beta_1^\top X, \beta_2^\top X, \ldots, \beta_d^\top X)$ of the $p$-dimensional predictor vector $X$.

Sufficient dimension reduction (SDR) is a commonly used approach for multi-index models. SDR focuses on finding a $p \times d$ matrix $\boldsymbol{\beta}$, such that

$$Y \perp\!\!\!\perp X \mid \boldsymbol{\beta}^\top X. \tag{1}$$

Equation (1) is equivalent to saying that the conditional distribution of $Y$ given $X$ is the same as that of $Y$ given $\boldsymbol{\beta}^\top X$, which is referred to as the regression information. Then $X$ can be replaced by

---

[*]Corresponding author

First Conference on Parsimony and Learning (CPAL 2024).

its $d$-dimensional representation $\boldsymbol{\beta}^\top X$ without losing any regression information of $Y$ on $X$. The matrix $\boldsymbol{\beta}$ satisfying (1) is not identifiable up to a multiplication by a full-rank matrix. For any full-rank matrix $A \in \mathbb{R}^{p \times p}$, it is easy to prove that $A\boldsymbol{\beta}$ also satisfies Equation (1). Fortunately, the space spanned by the columns of $\boldsymbol{\beta}$, called dimension reduction subspace (DRS) denoted as $\mathrm{Span}(\boldsymbol{\beta})$, is identifiable. However, DRS is not unique. For example, $\mathbb{R}^p$ is a trivial DRS because $Y \perp\!\!\!\perp X \mid I_p^\top X$ holds, where $I_p$ is the $p \times p$ identity matrix and $\mathrm{Span}(I_p) = \mathbb{R}^p$. Therefore, the main interest is to estimate the central subspace $\mathcal{S}_{Y|X} \colon \mathcal{S}_{Y|X} = \bigcap_{\boldsymbol{\beta}} \mathrm{Span}(\boldsymbol{\beta})$, which is the intersection of all DRSs if itself is a DRS. Under mild conditions [4, 5], the central subspace exists and is unique. The ultimate goal is to find one basis of the central subspace. When the response variable $Y$ is in a Euclidean space, numerous methods have been developed for SDR, including sliced inverse regression (SIR) [6], sliced average variance estimation [7], principal Hessian direction [8], cumulative mean estimation [9], fused inverse regression estimators [10], and the Fourier transform approach [11]. See related chapters in [12] for a comprehensive discussion. In particular, some dimension reduction methods, such as the minimum discrepancy approach [10, 13, 14], decompose the kernel matrix into two low-rank matrices. This decomposition can be viewed as a form of sparse dictionary learning [15].

Classical SDR methods are not readily applicable to non-Euclidean responses. For illustration, consider the application in the context of image completion for handwritten digits [16]. Here, a portion of each image serves as the response, situated on a manifold equipped with a distance metric, while the remaining part of the image is employed as predictors in a high-dimensional Euclidean space. Recent advances in Fréchet SDR have enabled its application to regression problems involving metric-space valued responses and Euclidean predictors [16]. One idea is to transform the Fréchet SDR into a traditional SDR by mapping the metric-space response to a real-valued response [17]. Another paper by [16] developed a weighted inverse regression ensemble (WIRE) approach that leverages the metric distance of random objects to construct a novel WIRE kernel matrix, which is used to estimate the central subspace. Nonetheless, when the number of features surpasses the sample size, the singularity of the covariance matrix often causes the failure of most kernel-based approaches.

Another challenge arose from high-dimensional Fréchet SDR problems is to allow sparsity in estimating the central subspace, as it is often assumed that only a few predictors significantly impact the response. Sufficient variable selection aims to find a subset of predictors, indexed by a set $S \in \{1, \ldots, p\}$, such that $Y \perp\!\!\!\perp X \mid X_S$, where $X_S = \{X_j, j \in S\}$. Since each row of the coefficient matrix $\boldsymbol{\beta}$ corresponds to one feature, nonzero rows of the matrix situate at the index set $S$. A widely adopted approach to identify influential variables in SDR is to formulate an optimization problem with regularization, as seen in [18–30], and many other works referenced within these sources. Most of the existing sparse SDR methods employed Lasso or group-Lasso penalty, both of which are convex and lead to biased estimation. Nonconvex penalization offers a promising alternative with reduced estimation bias. Some commonly used nonconvex penalties include the minimax concave penalty (MCP) [31], the smooth clipped absolute deviation (SCAD) [32], and the capped $\ell_1$-penalty [33]. However, nonconvex optimization can be computationally and analytically challenging as it gives rise to the issue of multiple local minima. To address this issue, a local linear approximation (LLA) is introduced by [34] for maximizing the nonconcave penalized likelihood. The authors employed a one-step LLA estimator that benefits from the oracle property of an initial estimator, but its performance is rather sensitive to the choice of the initial start.

**Contributions.** To bridge the existing research gaps, we propose a novel approach for high-dimensional Fréchet SDR by augmenting the WIRE technique with a nonconvex penalty. Our main contributions are as follows: (1) We propose a multitask regression approach to estimate the central subspace of Fréchet regression. The essential advantage of this procedure is that it eliminates the need for the inverse of the large covariance matrix, which is the main impediment of using kernel-based prototypes in high-dimensional settings. (2) We achieve simultaneous variable selection by incorporating a nonconvex penalty into the multitask regression framework. This inclusion enables the selection of relevant variables and helps mitigate estimation bias. (3) We develop a scalable optimization algorithm, called double approximation shrinkage-thresholding, to implement the pro-

posed method. The double approximation technique is less sensitive to initial values compared to the one-step LLA procedure. Our proposed algorithm provides explicit expressions at each iteration, thereby enhancing computational efficiency and convergence properties. The Python code for generating all of the experimental figures in this paper can be accessed at: [code repository].

## 2. Methodology

### 2.1. Preliminary

Let $(\Omega, m)$ be a metric space, where $m(\cdot, \cdot)$ is a metric from $\Omega \times \Omega \to \mathbb{R}$. We consider the response variable $Y \in \Omega$ and the predictors $X \in \mathbb{R}^p$. Unlike linear regression, which considers $\mathbb{E}(Y \mid X)$, a SIR procedure proposed by [6] estimates the inverse regression $\mathbb{E}(X \mid Y)$ using intervals of $Y$. The SIR approach, similar to other kernel-based SDR methods, utilizes singular value decomposition on the SIR kernel matrix $\Sigma^{-1} \text{var}[\mathbb{E}(X \mid Y)]$ to recover the central subspace, where $\Sigma = \text{var}(X)$.

To introduce our method, we first review the martingale difference divergence matrix (MDDM) proposed by [35], where MDDM is utilized to define the martingale difference correlation, quantifying the departure from conditional mean independence between two random vectors. The definition of MDDM is as follows:

$$-\mathbb{E}[\{Y - \mathbb{E}(Y)\}^\top \{\widetilde{Y} - \mathbb{E}(\widetilde{Y})\} \| X - \widetilde{X} \|],$$

where $(\widetilde{X}, \widetilde{Y})$ is an independent copy of $(X, Y)$, and both are random vectors.

While our primary focus is not on studying independence, understanding the origin of the WIRE kernel matrix is crucial. Instead of assessing independence, recent research papers by [16, 30] applied the concept of inverse regression to define a WIRE matrix, switching the roles of response and predictor vectors within the framework of the MDDM to handle random objects in metric space. The corresponding WIRE kernel matrix is defined as follows:

$$\Lambda = -\mathbb{E}\{(X - \mu)(\widetilde{X} - \mu)^\top m(Y, \widetilde{Y})\} \in \mathbb{R}^{p \times p}, \tag{2}$$
$$= -\mathbb{E}[\mathbb{E}\{X - \mu \mid Y\} \mathbb{E}\{\widetilde{X} - \mu \mid Y\}^\top m(Y, \widetilde{Y})],$$

where the weight $m(Y, \widetilde{Y})$ is the metric distance between $Y$ and $\widetilde{Y}$ and $\mu = \mathbb{E}(X)$. Therefore, $\Lambda$ in (2) could be expressed as the weighted average ensemble of the inverse regression means.

**Definition 1.** (*Negative Type;* [36]) *The semimetric space* $(\Omega, m)$ *is said to have a negative type if, for any* $n \geq 2$, $y_1, \ldots, y_n \in \Omega$, *and* $a_1, \ldots a_n \in \mathbb{R}$, *with* $\sum_{i=1}^n a_i = 0$:

$$\sum_{i=1}^n \sum_{j=1}^n a_i a_j m(y_i, y_j) \leq 0.$$

**Proposition 1.** (*Proposition 1 in* [16]) *Assume that* $(\Omega, m)$ *is a metric space of negative type, and the linearity condition holds, that is,* $\mathbb{E}(X \mid \boldsymbol{\beta}^\top X)$ *is linear in* $X$, *then* $\Lambda$ *is positive semidefinite and* $\text{Span}\{\Sigma^{-1} \Lambda\} \subseteq \mathcal{S}_{Y|X}$.

The metric space of negative type ensures that the kernel matrix $\Lambda$ is a positive semidefinite matrix. Examples of distributions satisfying the linearity condition include the normal distribution and the elliptical distributions. Based on Proposition 1, we can perform a singular decomposition on $\Sigma^{-1} \Lambda$ to obtain the left eigenvectors, which serve as one basis for the central subspace.

Let $\boldsymbol{\beta} = (\beta_1, \ldots, \beta_d)$ be the left singular vectors of $\Sigma^{-1} \Lambda$ corresponding to the $d$ largest singular values $(\psi_1, \ldots, \psi_d)$. If we further assume that the coverage condition holds, that is, $\text{Span}\{\Sigma^{-1} \Lambda\} = \mathcal{S}_{Y|X}$, then $\boldsymbol{\beta}$ provides a basis of $\mathcal{S}_{Y|X}$. Thus, we have

$$\Sigma \text{Span}(\boldsymbol{\beta}) = \text{Span}(\Lambda). \tag{3}$$

In the process of estimating a basis for the central subspace, the initial step involves substituting a sample estimate of $\Sigma^{-1} \Lambda$, which requires calculating the inverse of the covariance matrix. However, estimating the precision matrix is a well-known challenge in the scenario where the predictor dimension significantly exceeds the sample size.

## 2.2. Sparse weighted inverse regression ensemble

Given data $\{(x_i, y_i), i = 1, \ldots, n\}$, estimate $\Sigma$ and $\Lambda$ by their sample counterparts $\widehat{\Sigma} = n^{-1} \sum_{i=1}^{n} (x_i - \bar{x})(x_i - \bar{x})^\top$, and $\widehat{\Lambda} = -\{n(n-1)\}^{-1} \sum_{1 \leq i \neq j \leq n} (x_i - \bar{x})(x_j - \bar{x})^\top m(y_i, y_j)$, where $\bar{x} = \frac{1}{n} \sum_{i=1}^{n} x_i$. Let $\mathbb{X} = (x_1 - \bar{x}, \ldots, x_n - \bar{x})^\top \in \mathbb{R}^{n \times p}$ and $D = (m(y_i, y_j))_{ij} \in \mathbb{R}^{n \times n}$.

A Lasso-SIR approach proposed by [26] estimates the central subspace by constructing synthetic response variables from the top eigenvectors of the SIR kernel matrix. Due to the essence of $\Lambda$ and its similar expression to the SIR kernel matrix, we consider performing eigenvalue decomposition on the matrix $\Lambda$. Let $\boldsymbol{\eta} = (\eta_1, \ldots, \eta_d)$ be the eigenvectors of $\Lambda$ corresponding to the $d$ largest eigenvalues $(\phi_1, \ldots, \phi_d)$. Due to the fact that $\Sigma \mathrm{Span}(\boldsymbol{\beta}) = \mathrm{Span}(\Lambda)$, we have $\boldsymbol{\eta} \propto \Sigma \boldsymbol{\beta}$.

Furthermore, we consider the sample estimates of $\widehat{\Lambda}$ and $\widehat{\Sigma}$, and denote the eigenvalues and eigenvectors of $\widehat{\Lambda}$ as $\hat{\phi}_1 \geq \hat{\phi}_2 \geq \ldots \geq \hat{\phi}_d$ and $\hat{\boldsymbol{\eta}} = (\hat{\eta}_1, \ldots, \hat{\eta}_d)$, respectively. According to the property of eigenvectors, we have $\hat{\boldsymbol{\eta}} \mathrm{diag}\{\hat{\phi}_1, \ldots, \hat{\phi}_d\} = \widehat{\Lambda} \hat{\boldsymbol{\eta}}$. After some algebra, $\widehat{\Lambda}$ can be written as $\widehat{\Lambda} = -\frac{1}{n(n-1)} \mathbb{X}^\top D \mathbb{X}$. Plug it into the equation of eigenvectors and eigenvalues, we have

$$\hat{\boldsymbol{\eta}} \mathrm{diag}\{\hat{\phi}_1, \ldots, \hat{\phi}_d\} = -\frac{1}{n(n-1)} \mathbb{X}^\top D \mathbb{X} \hat{\boldsymbol{\eta}}.$$

**Definition 2.** *The synthetic response variables for sparse WIRE are defined as follows:*

$$\mathbb{Y} = -\frac{1}{n-1} D \mathbb{X} \hat{\boldsymbol{\eta}} \mathrm{diag}\{1/\hat{\phi}_1, \ldots, 1/\hat{\phi}_d\}.$$

Based on the construction of $\mathbb{Y}$, we have $\hat{\boldsymbol{\eta}} = \frac{1}{n} \mathbb{X}^\top \mathbb{Y}$.

When employing sample estimates to approximate $\Sigma$ and $\boldsymbol{\eta}$, $\boldsymbol{\eta} \propto \Sigma \boldsymbol{\beta}$ can be written as $\hat{\boldsymbol{\eta}} = \frac{1}{n} \mathbb{X}^\top \mathbb{Y} \propto \frac{1}{n} \mathbb{X}^\top \mathbb{X} \boldsymbol{\beta}$. This aligns precisely with the normal equation in the multiple regression model. Define a function $f(\boldsymbol{\beta}) = \frac{1}{2n} \|\mathbb{Y} - \mathbb{X}\boldsymbol{\beta}\|_{\mathrm{F}}^2$. The gradient of $f(\boldsymbol{\beta})$ is $\nabla f(\boldsymbol{\beta}) = -\frac{1}{n} \mathbb{X}^\top (\mathbb{Y} - \mathbb{X}\boldsymbol{\beta})$. Set it to zero, then we obtain the normal equation $\frac{1}{n} \mathbb{X}^\top \mathbb{X} \boldsymbol{\beta} = \frac{1}{n} \mathbb{X}^\top \mathbb{Y}$. The expression motivates us to conduct a regression analysis, regressing the $d$-dimensional responses $\mathbb{Y}$ on the $p$-dimensional predictor $\mathbb{X}$.

Based on previous discussions, we are ready to introduce a penalized multitask regression approach [37] for recovering the central subspace $\mathcal{S}_{Y|X} = \mathrm{Span}\{\boldsymbol{\beta}\}$. The objective function is as follows:

$$L(\boldsymbol{\beta}) = \frac{1}{2n} \|\mathbb{Y} - \mathbb{X}\boldsymbol{\beta}\|_{\mathrm{F}}^2 + g(\boldsymbol{\beta}), \tag{4}$$

where $g(\boldsymbol{\beta})$ is a sparsity-inducing penalty term, and $\|\cdot\|_{\mathrm{F}}$ is the Frobenius norm. Formulating the problem as a regression offers advantages, one being the elimination of the need to estimate the inverse of $\Sigma$. This is especially beneficial when $n < p$ since $\hat{\Sigma}$ can be singular. Additionally, regression problems with penalization have been extensively studied, and several penalties can be employed in this context, such as two well-researched penalties: Lasso and group Lasso.

## 2.3. Sparse WIRE with minimax concave penalty

Convex penalties, like Lasso and group Lasso, induce sparsity and select variables, but these approaches suffer from underestimation bias in their estimates. The estimation bias severely interferes with variable selection when $p$ and the number of active variables $|S|$ are both large. Some commonly used nonconvex penalties have been developed to overcome this bias, including the SCAD [32] and MCP [31]. Both the SCAD penalty, that is $\lambda \int_0^t \min\{1, (\gamma - x/\lambda)_+/(\gamma - 1)\}dx, \gamma > 0$, and MCP defined on $[0, \infty)$,

$$\rho(|t|; \lambda) = \begin{cases} \lambda|t| - t^2/(2\gamma), & \text{if } 0 \leq |t| < \lambda\gamma; \\ \lambda^2\gamma/2, & \text{if } |t| \geq \lambda\gamma, \end{cases}$$

mitigate the bias of estimators by choosing a constant penalty beyond a threshold level $\lambda\gamma$. Here, $\lambda > 0$ and $\gamma > 0$ are regularization parameters. While the main focus of this paper lies in utilizing the

MCP for illustrative purposes, all discussions presented here can be extended to other nonconvex penalties as well. Without confusion, $\rho(t)$ is used as $\rho(t;\lambda)$ for short. The advantage of the chosen $\rho(\cdot)$ is to achieve sparsity for those with small magnitude in $\boldsymbol{\beta}$ while keeping the original scale for the other large values.

The optimization problem with the Lasso penalty achieves entry-wide sparsity by applying the MCP to each entry of $\boldsymbol{\beta}$, that is,

$$\hat{\boldsymbol{\beta}} = \arg\min_{\boldsymbol{\beta} \in \mathbb{R}^{p \times d}} \frac{1}{2n} \|\mathbb{Y} - \mathbb{X}\boldsymbol{\beta}\|_{\mathrm{F}}^2 + \sum_{j=1}^{d} \sum_{i=1}^{p} \rho(|\boldsymbol{\beta}_{ij}|), \tag{5}$$

where $\boldsymbol{\beta}_{ij}$ is the $(i, j)$-entry of $\boldsymbol{\beta}$. To achieve a row sparse estimator $\boldsymbol{\beta}$, we implement $\rho(\cdot)$ on each row of $\boldsymbol{\beta}$, that is to choose $g(\boldsymbol{\beta}) = \sum_{i=1}^{p} \rho(\|e_i^\top \boldsymbol{\beta}\|_2)$, where $e_i$ is the $i$th canonical basis vector in $\mathbb{R}^p$. So the optimization problem (6) equipped with a group-wise MCP is,

$$\hat{\boldsymbol{\beta}} = \arg\min_{\boldsymbol{\beta} \in \mathbb{R}^{p \times d}} \frac{1}{2n} \|\mathbb{Y} - \mathbb{X}\boldsymbol{\beta}\|_{\mathrm{F}}^2 + \sum_{i=1}^{p} \rho(\|e_i^\top \boldsymbol{\beta}\|_2). \tag{6}$$

The nonconvexity of a minimization problem results in many local minima and is the main cause of computational and analytical difficulties. An LLA algorithm is introduced by [34] for maximizing the nonconcave penalized likelihood function. The key idea of the algorithm involves initially constructing a linear approximation of the penalty term and subsequently solving the updated optimization problem through iteration. Following the same reasoning as Theorem 2 in [34], we can prove that the linear approximation of $\rho(\cdot)$ is the best convex function of $\rho(\cdot)$ as stated in the following lemma.

**Lemma 1.** *Let $\rho^*(t \mid t_0) = \rho(t_0) + (t - t_0)\rho'(t_0)$ be the linear approximation of $\rho(\cdot)$ at $t_0$, and $\rho'(t) = \max\{\lambda - t/\gamma, 0\}$, for $t > 0$. Suppose that $\phi(\cdot)$ is any convex majorization function of $\rho(\cdot)$ at $t_0$, that is*

$$\phi(t_0) = \rho(t_0) \quad and \quad \phi(t) \geq \rho(t), \quad for\ any\ t.$$

*Then $\rho^*(t \mid t_0)$ is a majorization function of $\rho(\cdot)$ at $t_0$, and $\phi(t) \geq \rho^*(t \mid t_0)$, for all $t$.*

The penalty $\rho(\cdot)$ is non-decreasing and has a continuous derivative in $(0, \infty)$. Also, $\rho'(0+) = \lambda > 0$, so the minimizers of (5) and (6) possess variable selection properties with certain entries set to zero [38].

## 3. Algorithm

### 3.1. Local linear approximation to penalty term

The LLA algorithm generates a sequence $\hat{\boldsymbol{\beta}}^{(k)}$ by minimizing a series of convex optimization problems:

$$\hat{\boldsymbol{\beta}}^{(k+1)} = \arg\min_{\boldsymbol{\beta} \in \mathbb{R}^{p \times d}} \frac{1}{2n} \|\mathbb{Y} - \mathbb{X}\boldsymbol{\beta}\|_{\mathrm{F}}^2 + \sum_{i=1}^{p} \rho^*(\|e_i^\top \boldsymbol{\beta}\|_2 \mid \|e_i^\top \boldsymbol{\beta}^{(k)}\|_2), \quad for\ k = 0, 1, 2, \ldots.$$

The iteration stops when the sequence $\{\boldsymbol{\beta}^{(k)}\}$ converges. After omitting constant terms with respect to $\boldsymbol{\beta}$, we only need to solve: $\arg\min_{\boldsymbol{\beta}} \frac{1}{2n} \|\mathbb{Y} - \mathbb{X}\boldsymbol{\beta}\|_{\mathrm{F}}^2 + \lambda \sum_{j=1}^{p} \|e_j^\top \boldsymbol{\beta}\|_2 \rho'(\|e_j^\top \boldsymbol{\beta}^{(k)}\|_2)$, which can be considered an adjusted group Lasso. Let $e_j^\top \boldsymbol{\beta}^* = \rho'(\|e_j^\top \boldsymbol{\beta}^{(k)}\|_2) e_j^\top \boldsymbol{\beta}$ and $X^* e_j = \frac{1}{\rho'(\|e_j^T \boldsymbol{\beta}^{(k)}\|_2)} \mathbb{X} e_j$. Then, it is equivalent to solve: $\arg\min_{\boldsymbol{\beta}^*} \frac{1}{2n} \|\mathbb{Y} - X^* \boldsymbol{\beta}^*\|_{\mathrm{F}}^2 + \lambda \sum_{j=1}^{p} \|e_j^\top \boldsymbol{\beta}^*\|_2$.

Define $G(\boldsymbol{\beta}) = \frac{1}{2n} \|\mathbb{Y} - \mathbb{X}\boldsymbol{\beta}\|_{\mathrm{F}}^2 + \sum_{i=1}^{p} \rho(\|e_i^\top \boldsymbol{\beta}\|_2)$ and $N(\boldsymbol{\beta}, \boldsymbol{\beta}') = \frac{1}{2n} \|\mathbb{Y} - \mathbb{X}\boldsymbol{\beta}\|_{\mathrm{F}}^2 + \sum_{i=1}^{p} \rho^*(\|e_i^\top \boldsymbol{\beta}\|_2 \mid \|e_i^\top \boldsymbol{\beta}'\|_2)$.

**Theorem 1.** *For a differentiable concave penalty function $\rho(\cdot)$ on $[0, \infty)$, we have*

$$G(\boldsymbol{\beta}) \leq N(\boldsymbol{\beta}, \boldsymbol{\beta}^{(k)}), \ for\ any\ \boldsymbol{\beta}, \quad and \quad G(\boldsymbol{\beta}^{(k)}) = N(\boldsymbol{\beta}^{(k)}, \boldsymbol{\beta}^{(k)}).$$

*Furthermore, the LLA algorithm has the descent property, that is, for $k \geq 0$, $G(\boldsymbol{\beta}^{(k+1)}) \leq G(\boldsymbol{\beta}^{(k)})$.*

Theorem 1 indicates that the sequence of function values $\{G(\boldsymbol{\beta}^{(k)})\}$ is nonincreasing given $\{\boldsymbol{\beta}^{(k)}\}$. Each iteration involves solving an optimization problem, which is computationally expensive. Hence, a one-step estimator approach is proposed by [34]. This approach firstly uses the ordinary least squared estimate as the initial value $\boldsymbol{\beta}_0$, and then obtains the final estimate $\boldsymbol{\beta}_1$ by iterating the LLA algorithm once. They have shown that the one-step LLA estimator shares the oracle property with good initial estimators when properly chosen parameters are used. We adopt their ideas and introduce a one-step LLA estimator for sparse WIRE. In particular, the initial $\boldsymbol{\beta}_0$ is chosen from the penalized multitask regression with the Lasso penalty term. The detailed algorithm is stated below.

---

**Algorithm 1:** One-step LLA Algorithm for group-wise MCP.

---

**Input:** $\gamma$.

**for** $i \in \{1, \ldots, d\}$ **do**

    1. Solve the multitask Lasso regression problem:
    $\hat{\boldsymbol{\alpha}} = \arg\min_{\boldsymbol{\alpha}} \frac{1}{2n}\|\mathbb{Y} - \mathbb{X}\boldsymbol{\alpha}\|_{\mathrm{F}}^2 + \lambda \sum_{j=1}^{p} \|e_j^\top \boldsymbol{\alpha}\|_2$, where $\lambda$ is selected using cross-validation.

    2. For $j \in \{1, \ldots, p\}$, define $X^* e_j = \frac{1}{w_j}\mathbb{X}e_j$, where $w_j = \rho'(\|e_j^\top \hat{\boldsymbol{\alpha}}\|_2) + \epsilon$ and $\epsilon = 2e^{-16}$.

    3. Solve the new regression again: $\hat{\boldsymbol{\beta}}^* = \arg\min_{\boldsymbol{\beta}} \frac{1}{2n}\|\mathbb{Y} - X^*\boldsymbol{\beta}\|_2^2 + \lambda \sum_{j=1}^{p}\|e_j^\top \boldsymbol{\beta}\|_2$, where $\lambda$ is selected using cross-validation.

    4. Each row of $\hat{\boldsymbol{\beta}}$ is updated by $e_j^\top \hat{\boldsymbol{\beta}}^* / w_j$.

**end**

---

From a computational point of view, $\rho'(\|e_j^\top \hat{\boldsymbol{\alpha}}\|_2)$ could be zero for some $j$. Hence, we add a small value to avoid division by zero: $w_j = \rho'(\|e_j^\top \hat{\boldsymbol{\alpha}}\|_2) + \epsilon$.

## 3.2. Double approximation shrinkage-thresholding algorithm

The one-step LLA estimator is highly sensitive to the initial value. To address this limitation, a naive approach is to apply the LLA algorithm to obtain a $k$-step estimator. However, this introduces additional complexity as each iteration requires solving a new penalized regression problem. We introduce a novel approach that combines a linear approximation to the penalty term with a quadratic approximation to the loss function.

When dealing with the unconstrained problem of a continuously differentiable function $f(\cdot)$: $\min\{f(\boldsymbol{\beta}) : \boldsymbol{\beta} \in \mathbb{R}^{p \times d}\}$, we generate a sequence $\{\boldsymbol{\beta}^{(k)}\}$ using the gradient iteration, a proximal regularization of the linearized function $f$ at $\boldsymbol{\beta}^{(k-1)}$,

$$\boldsymbol{\beta}^{(k)} = \arg\min_{\boldsymbol{\beta} \in \mathbb{R}^{p \times d}} \left\{ f(\boldsymbol{\beta}^{(k-1)}) + \mathrm{Tr}[(\boldsymbol{\beta} - \boldsymbol{\beta}^{(k-1)})^\top \nabla f(\boldsymbol{\beta}^{(k-1)})] + \frac{1}{2t_k}\|\boldsymbol{\beta} - \boldsymbol{\beta}^{(k-1)}\|_{\mathrm{F}}^2 \right\},$$

where $t_k$ is a step size, $\nabla f(\boldsymbol{\beta}) = -\frac{1}{n}\mathbb{X}^\top(\mathbb{Y} - \mathbb{X}\boldsymbol{\beta})$ is the gradient of $f(\boldsymbol{\beta})$, and $\mathrm{Tr}(A)$ is the trace of matrix $A$. Then, we apply the same gradient idea in the penalized optimization problem, that is

$$\boldsymbol{\beta}^{(k)} = \arg\min_{\boldsymbol{\beta} \in \mathbb{R}^{p \times d}} \left\{ f(\boldsymbol{\beta}^{(k-1)}) + \mathrm{Tr}[(\boldsymbol{\beta} - \boldsymbol{\beta}^{(k-1)})^\top \nabla f(\boldsymbol{\beta}^{(k-1)})] + \frac{1}{2t_k}\|\boldsymbol{\beta} - \boldsymbol{\beta}^{(k-1)}\|_{\mathrm{F}}^2 + \sum_{i=1}^{p} \rho^*(\|e_i^\top \boldsymbol{\beta}\|_2 \mid \|e_i^\top \boldsymbol{\beta}^{(k-1)}\|_2) \right\}$$

$$= \arg\min_{\boldsymbol{\beta} \in \mathbb{R}^{p \times d}} \left\{ \frac{1}{2t_k}\|\boldsymbol{\beta} - (\boldsymbol{\beta}^{(k-1)} - t_k \nabla f(\boldsymbol{\beta}^{(k-1)}))\|_{\mathrm{F}}^2 + \sum_{i=1}^{p} \rho'(\|e_i^\top \boldsymbol{\beta}^{(k-1)}\|_2)\|e_i^\top \boldsymbol{\beta}\|_2 \right\}.$$

The second equation rewrites the first one after ignoring constant terms. Solving $\boldsymbol{\beta}^{(k)}$ reduces to a $\ell_2$ penalization for each row of matrix $\boldsymbol{\beta}^{(k)}$. We use the block soft-thresholding operator $S_\lambda(a) = \max(1 - \frac{\lambda}{\|a\|}, 0)a$ to obtain each row: $e_i^\top \boldsymbol{\beta}^{(k)} = S_{t_k \rho'(\|e_i^\top \boldsymbol{\beta}^{(k-1)}\|_2)}(e_i^\top \boldsymbol{\beta}^{(k-1)} - t_k e_i^\top \nabla f(\boldsymbol{\beta}^{(k-1)}))$. Motivated by [39], choose $t_k$ to be $1/L$, where $L$ is the smallest Lipschitz constant of the gradient $\nabla f$ and $L = \|\frac{1}{n}\mathbb{X}^\top \mathbb{X}\|_2$.

Define a quadratic function of $\boldsymbol{\alpha}$ given $\boldsymbol{\beta}$,

$$f^*(\boldsymbol{\alpha}, \boldsymbol{\beta}) = f(\boldsymbol{\beta}) + \mathrm{Tr}[(\boldsymbol{\alpha} - \boldsymbol{\beta})^\top \nabla f(\boldsymbol{\beta})] + \frac{L}{2}\|\boldsymbol{\alpha} - \boldsymbol{\beta}\|_{\mathrm{F}}^2.$$

It is straightforward to show that $f(\boldsymbol{\alpha}) \leq f^*(\boldsymbol{\alpha}, \boldsymbol{\beta})$ for every $\boldsymbol{\alpha}, \boldsymbol{\beta} \in \mathbb{R}^{p \times d}$ and $f(\boldsymbol{\beta}) = f^*(\boldsymbol{\beta}, \boldsymbol{\beta})$. Hence, $f^*(\boldsymbol{\alpha}, \boldsymbol{\beta})$ is surrogate function of $f(\boldsymbol{\alpha})$.

**Theorem 2.** *For each $k \geq 1$, $\boldsymbol{\beta}^{(k)}$ is a minimizer of $M(\boldsymbol{\beta}, \boldsymbol{\beta}^{(k-1)}) = f^*(\boldsymbol{\beta}, \boldsymbol{\beta}^{(k-1)}) + \sum_{i=1}^{p} \rho^*(\|e_i^\top \boldsymbol{\beta}\|_2 \mid \|e_i^\top \boldsymbol{\beta}^{(k-1)}\|_2)$, then we have*

$$G(\boldsymbol{\beta}) \leq M(\boldsymbol{\beta}, \boldsymbol{\beta}^{(k-1)}), \text{ for any } \boldsymbol{\beta}, \quad and \quad G(\boldsymbol{\beta}^{(k-1)}) = M(\boldsymbol{\beta}^{(k-1)}, \boldsymbol{\beta}^{(k-1)}).$$

*Furthermore, for $k \geq 1$, $G(\boldsymbol{\beta}^{(k)}) \leq G(\boldsymbol{\beta}^{(k-1)})$.*

Theorem 2 shows that the sequence of function values $\{G(\boldsymbol{\beta}^{(k)})\}$ is noninceasing, signifying consistent improvement after each iteration. This algorithm, iterative minimizing $M(\boldsymbol{\beta}, \boldsymbol{\beta}^{(k-1)})$, is called the double approximation shrinkage-thresholding algorithm, denoted as DASTA, since the objective function $M$ utilizes local approximation functions for $f$ and $\rho$ simultaneously.

---

**Algorithm 2:** DASTA for group-wise MCP.

---

**Input:** $\lambda$ and $\gamma$.
**Initialize** the algorithm with $\boldsymbol{\beta}_0$ and set $k = 0$.
**repeat**

> 1. Calculate $O^{(k+1)} = \boldsymbol{\beta}^{(k)} - \frac{1}{L}\nabla f(\boldsymbol{\beta}^{(k)}) = \boldsymbol{\beta}^{(k)} + \frac{1}{nL}\mathbb{X}^\top(\mathbb{Y} - \mathbb{X}\boldsymbol{\beta}^{(k)})$.
>
> 2. Calculate $w_i^{(k+1)} = \rho'(\|e_i^\top \boldsymbol{\beta}^{(k)}\|_2)/L$, for $i \in \{1, \dots, p\}$.
>
> 3. The $i$-row of $\boldsymbol{\beta}^{(k+1)}$ is $S_{w_i^{(k+1)}}(e_i^\top O^{(k+1)})$.
>
> 4. $k = k + 1$.

**until** $\|\boldsymbol{\beta}^{(k+1)} - \boldsymbol{\beta}^{(k)}\|_{\mathrm{F}} \leq \epsilon$;

---

Unlike the LLA algorithm, which involves solving a new penalized regression problem in each iteration, the DASTA algorithm simplifies each iteration by leveraging matrix operations to derive closed-form solutions. This results in significant computational benefits, making this algorithm computationally efficient.

## 3.3. Implementation details

Both Algorithms 1 and 2 necessitate the value of the structural dimension $d$, even though it might be unknown in practical scenarios. In [26], the authors have proved that there is a disparity among the adjusted eigenvalues $\Phi_i = \hat{\phi}_i \|e_i^\top \hat{\boldsymbol{\beta}}\|_2$ before and after the actual dimension $d$. Therefore, they applied the K-means clustering on $\Phi_i$ with two groups. The estimated $d$ is the number of points in the group with larger $\Phi_i$ values. Inspired by their findings, we employ the same algorithm on the estimated $\hat{\boldsymbol{\beta}}$ to ascertain the appropriate value of $d$ for each algorithm.

---

**Algorithm 3:** Determine structural dimension $d$.

---

> 1. Estimate $\hat{\boldsymbol{\beta}}$ when $d = m$ using one of Algorithms 1 and 2, where $m$ is large enough number. For example, $m = 10$.
>
> 2. For $i \in \{1, \dots, m\}$, calculate $\Phi_i = \hat{\phi}_i \|e_i^\top \hat{\boldsymbol{\beta}}\|_2$.
>
> 3. Apply K-means clustering on $\{\Phi_i\}$ with $K = 2$.
>
> 4. The estimated value of $\hat{d}$ is the total number in the cluster with larger $\{\Phi_i\}$.

---

When dealing with nonconvex penalization, we fix $\gamma$ to be 2 and then utilize cross-validation to determine an optimal $\lambda$. The choice of $\lambda$ exhibits high robustness. Our simulation study only shows results when $\lambda_{\max}/6$, where $\lambda_{\max} = \max\{|\mathbb{X}^\top \mathbb{Y}/n|_{ij}\}$ or $\max\{\|e_i^\top \mathbb{X}^\top \mathbb{Y}/n\|_2, i = 1, \dots, p\}$ for LLA and DASTA algorithms, respectively.

# 4. Numerical studies

## 4.1. Synthetic data analysis

To showcase the effectiveness of our proposed algorithms, we consider six methods: the LASSO and Group-LASSO estimators (See more details in Appendix A.2); two one-step estimators denoted as LLA_E and LLA_G by solving Equations (5) and (6); two DASTA estimators $DASTA_0$ and $DASTA_g$ using two initial values $\beta_0 = \mathbf{0}$ and the group Lasso estimate, respectively. Let $\widehat{S}$ be the index set of nonzero rows of $\widehat{\beta}$. To assess the performance, we utilize the following criteria: (1) General loss $\|\widehat{\beta}\widehat{\beta}^\top - \beta\beta^\top\|_F$, where $\beta$ and $\widehat{\beta}$ are normalized, (2) False positive, the number of inactive variables incorrectly indicated as active $|\widehat{S} \cap S^c|$, (3) False negative, the number of active variables incorrectly indicated as inactive $|S \cap \widehat{S}^c|$, and (4) Estimated structural dimension $\hat{d}$ using Algorithm 3. Let $X$ be a $p$-dimensional multivariate normal random vector: $X \sim N_p(0, \Sigma)$. We consider two covariance matrices: an identity matrix $\Sigma_1 = I_p$ and a Toeplitz matrix $\Sigma_2 = (0.5^{|i-j|})$. All three examples are multiple index models with $d = 2$. Let $\beta_1 = e_1 + \cdots + e_5$ and $\beta_2 = e_6 + \cdots + e_{10}$.

**Example 1.** *This example is multivariate responses with Euclidean metric.*
$$Y_1 = 1 + \beta_1^\top X + \epsilon_1, \quad Y_2 = \beta_2^\top X + \epsilon_2, \quad Y_3 = |\beta_1^\top X|\epsilon_3,$$
*where $\epsilon_1, \epsilon_2, \epsilon_3 \sim N(0,1)$ and are independent of each other.*

**Example 2.** *The response $Y$ is generated as the distribution with quantile function $Q_Y(\tau) = \beta_1^\top X + \beta_2^\top X \Phi^{-1}(\tau)$[16], where $\Phi(\cdot)$ is the cumulative distribution function of standard normal. The 2-Wasserstein distance is utilized to quantify the distance between two distributions.*

**Example 3.** *Consider the unit-sphere data [40], where the response $Y$ lives in a 3-dimensional unit sphere equipped with the geodesic distance $\arccos(Y^\top \widetilde{Y})$:*
$$Y_1 = \cos(\epsilon) \sin\{\beta_1^\top(X + \mathbf{1}_p)\} \sin\{\beta_2^\top(X + \mathbf{1}_p)\},$$
$$Y_2 = \cos(\epsilon) \sin\{\beta_1^\top(X + \mathbf{1}_p)\} \cos\{\beta_2^\top(X + \mathbf{1}_p)\},$$
$$Y_3 = \cos(\epsilon) \cos\{\beta_1^\top(X + \mathbf{1}_p)\}, \quad Y_4 = \sin(\epsilon),$$
*where $\epsilon \sim N(0, 0.1^2)$, and $\mathbf{1}_p$ is a all-ones vector with length $p$.*

Figure 1 in Appendix A.1 highlights the superior performance of nonconvex penalties in comparison to their convex counterparts for Examples 1 to 3. Specifically, all LLA and DASTA estimators using nonconvex penalties outperform LASSO and Group-LASSO, effectively reducing estimation bias and resulting in lower general losses. LLA_E displays a larger variation than DASTA estimators, as depicted in Figures 1(a,b,c). For Example 3 with $\Sigma_2$, LLA_E exhibits the highest loss among all six estimators, while the DASTA estimators produce competitive estimates, shown in Figure 1(f). Moreover, the DASTA algorithm provides similar estimates for different start values. Overall, the DASTA estimators show superior performance across different types of responses, including multivariate, distributed, and sphere data.

Figure 2 in Appendix A.1 illustrates the line plots for each method as the sample size $n$ varies from 300 to 2700, with a fixed value of $p = 3000$ and covariance matrix $\Sigma_1$. When the sample size is small, particularly at $n = 300$, LLA_E has the highest general loss in Examples 1 and 3. This high loss indicates the instability of the estimator when confronted with a small sample size, primarily due to an inaccurate initial value. On the other hand, in scenarios with insufficient sample size, the DASTA methods exhibit smaller loss values compared to LLA estimators.

The simulation results for variable selection and structural dimension detection when $n = 1000$ and $p = 2000$ are presented in Table 1 (see Appendix A.1). All six methods successfully identify all active variables without any false positives. However, the LASSO and Group-LASSO estimators tend to select more inactive variables as active, whereas the DASTA estimators achieve nearly perfect variable selection with smaller false positives and false negatives. Regarding the performance in detecting the structural dimension, the DASTA approaches consistently choose $\hat{d} = 2$ and outperform the other methods including the LLA estimators in various examples and settings.

## 4.2. Real data analysis: human mortality data

This section applies three methods: Group-LASSO, LLA_G, and DASTA$_g$ to analyze human mortality data[2]. The database recorded the number of deaths at five-year intervals for every country in 2021 from 0 to 100 years. These mortality distributions are considered as responses, and the Hellinger distance is used to quantify the distance between distributions. This section considers the high-dimensional regime where the number of predictors exceeds the sample size. We simultaneously perform sufficient dimension reduction and variable selection using the sparse Fréchet SDR approach. To this end, we use 236 countries and 52 demographic indicators (except for the Population Annual Doubling Time due to missing values) with additional 200 independent predictors drawn from the standard normal distribution. We then standardize all predictors by centering them at zero and scaling to unit variance.

The DASTA algorithm estimates the structural dimension of the central subspace to be one, while the other two methods choose a larger dimension. Therefore, we report the results for $\hat{d} = 1$. Table 2 in Appendix A.1 shows that Group-LASSO selects seven variables. On the contrary, LLA_G and DASTA$_g$ select a smaller set of variables. Notably, none of the methods selects any of the 200 noise predictors. The three estimates assign high loading values to male life expectancy at birth and female life expectancy at age 65. Other predictors have relatively small loading values, indicating a weaker impact on age-at-death distributions. To assess the information captured by our proposed methods, a 3D plot based on the DASTA$_g$ results is generated. Figure 3 in Appendix A.1 showcases the mortality densities and their maximum values against the first sufficient predictor $\widehat{\boldsymbol{\beta}}^\top X$, showing a positive relationship between the first sufficient predictor and longevity. These findings offer a fresh perspective on studying human mortality densities across the age range of 0 to 100 years. Consequently, policymakers, government entities, or healthcare workers can focus on implementing preventive measures against accidents or diseases and providing support for infants and elderly individuals around the age of 65. These efforts aim to promote healthier and longer lives for the population.

## 5. Conclusion

In this paper, we studied the sparse Fréchet SDR problem for regression of non-Euclidean responses on high-dimensional features. The sparse Fréchet SDR approach has great potential in representation learning with applications in image and video compression, natural language processing, recommendation systems, and financial analysis. We proposed a penalized multitask regression approach to recover the central subspace that contains the core information about the response. This strategy seamlessly facilitates the integration of various existing regularization techniques, such as LASSO, group LASSO, and MCP. The approach also avoids inverting large covariance matrices, a task that can often be computationally prohibitive. Notably, the method is amenable to a wide range of non-Euclidean response types, providing flexibility in modeling diverse data structures. We introduced a novel algorithm called DASTA. The DASTA algorithm effectively addresses the challenges posed by nonconvex optimization problems by transforming them into convex problems that can be solved efficiently. By doing so, it preserves the advantageous properties of nonconvex penalties, thereby eliminating estimation biases and enhancing variable selection. While our simulation results demonstrate the efficacy of the proposed methods, one of our future endeavors involves investigating the statistical properties, such as consistency and the minimax rate, of the sparse WIRE estimators derived from multitask regression with LASSO penalties. We expect that a similar convergence rate of $\sqrt{s \log p/n}$ can be achieved. Additionally, it would be interesting to investigate the statistical properties of MCP in this context.

---

[2]The data set can be downloaded from the World Population Prospects Database https://population.un.org/wpp/Download.

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

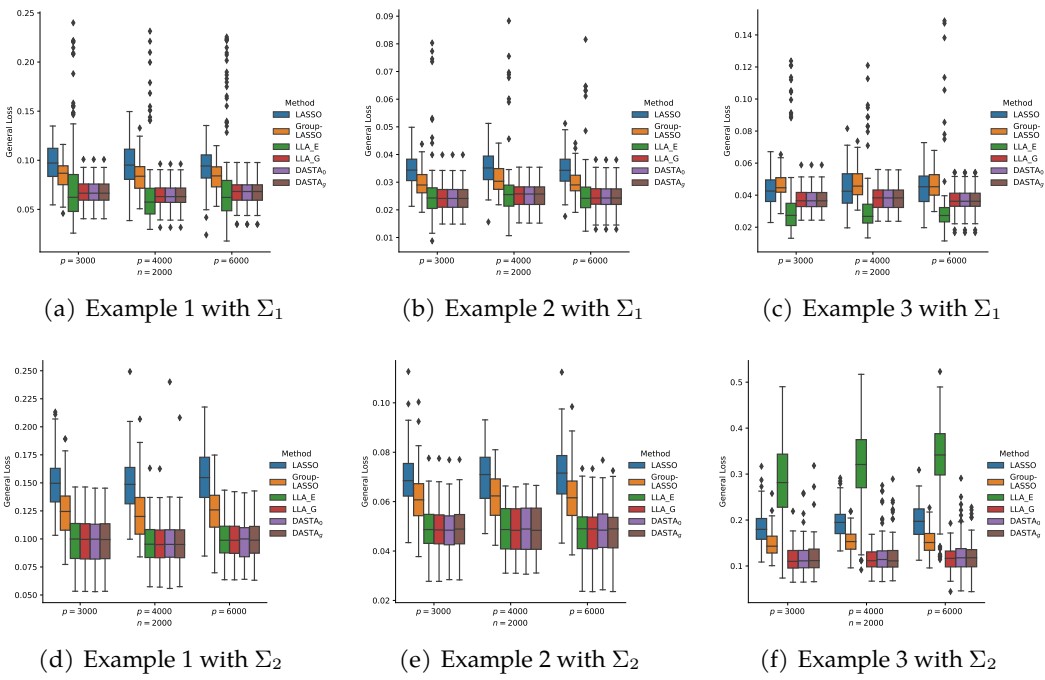

Figure 1: These figures show boxplots of general loss to compare six different methods: LASSO, Group-LASSO, LLA_E (LLA algorithm with entry-wise MCP), LLA_G (LLA algorithm with group-wise MCP), DASTA$_0$ (DASTA algorithm with initial value $\boldsymbol{\beta}_0 = 0$), and DASTA$_g$ (DASTA algorithm with the initial value from group Lasso estimate) for Examples 1–3 with covariance matrices $\Sigma_1$, $\Sigma_2$ and three different settings of $p = \{3000, 4000, 6000\}$, while keeping $n$ fixed at 2000.

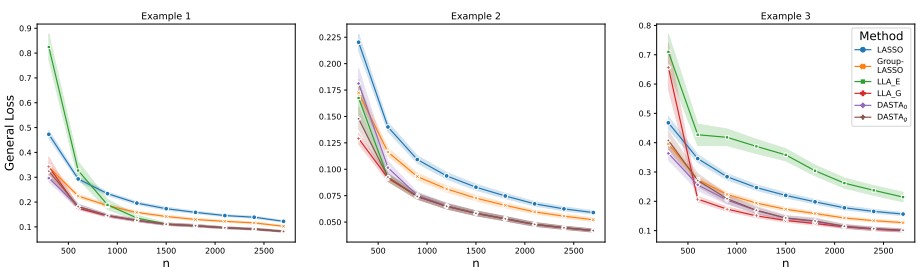

Figure 2: Line plots show the mean and 95% confidence intervals for six methods: LASSO, Group-LASSO, LLA_E, LLA_G, DASTA$_0$, and DASTA$_g$, with covariance matrix $\Sigma_1$ in Examples 1–3 when $p = 3000$.

# A. Appendix

## A.1. Simulation and real data results

Figure 1 highlights the superior performance of nonconvex penalties in comparison to their convex counterparts for Examples 1 to 3. Figure 2 illustrates the line plots for each method as the sample size $n$ varies from 300 to 2700, with a fixed value of $p = 3000$ and covariance matrix $\Sigma_1$. Table 1 shows the simulation results for variable selection and structural dimension detection when $n = 1000$ and $p = 2000$ for Examples 1 to 3.

Table 1: Compare six distinct methods: LASSO, Group-LASSO, LLA_E (LLA algorithm with entry-wise MCP), LLA_G (LLA algorithm with group-wise MCP), $\text{DASTA}_0$ (DASTA algorithm with initial value $\boldsymbol{\beta}_0 = 0$), and $\text{DASTA}_g$ (DASTA algorithm with the initial value from group Lasso estimate) and present the mean and standard deviation of false positive (FP), false negative (FN), and estimated $d$ based on 100 simulations for Examples 1 to 3 with two covariance structures when $n = 1000, p = 2000$.

| Cov | Example | | LASSO | Group-LASSO | LLA_E | LLA_G | $\text{DASTA}_0$ | $\text{DASTA}_g$ |
|---|---|---|---|---|---|---|---|---|
| $\Sigma_1$ | 1 | FP | 61.04(28.49) | 35.83(20.31) | 0.00(0.00) | 0.00(0.00) | 0.01(0.10) | 0.01(0.10) |
| | | FN | 0.00(0.00) | 0.00(0.00) | 0.00(0.00) | 0.00(0.00) | 0.00(0.00) | 0.00(0.00) |
| | | $\hat{d}$ | 2.00(0.00) | 4.10(3.65) | 2.00(0.00) | 3.99(3.58) | 2.00(0.00) | 2.00(0.00) |
| | 2 | FP | 62.09(25.84) | 31.38(19.24) | 0.00(0.00) | 0.00(0.00) | 0.00(0.00) | 0.00(0.00) |
| | | FN | 0.00(0.00) | 0.00(0.00) | 0.00(0.00) | 0.00(0.00) | 0.00(0.00) | 0.00(0.00) |
| | | $\hat{d}$ | 2.00(0.00) | 2.81(2.49) | 2.00(0.00) | 2.72(2.39) | 2.00(0.00) | 2.00(0.00) |
| | 3 | FP | 35.63(19.64) | 25.11(13.29) | 0.00(0.00) | 0.00(0.00) | 0.00(0.00) | 0.00(0.00) |
| | | FN | 0.00(0.00) | 0.00(0.00) | 0.00(0.00) | 0.00(0.00) | 0.00(0.00) | 0.00(0.00) |
| | | $\hat{d}$ | 2.00(0.00) | 2.08(0.08) | 2.00(0.00) | 2.08(0.08) | 2.00(0.00) | 2.00(0.00) |
| $\Sigma_2$ | 1 | FP | 40.59(23.10) | 19.94(17.06) | 0.00(0.00) | 0.00(0.00) | 0.00(0.00) | 0.00(0.00) |
| | | FN | 0.00(0.00) | 0.00(0.00) | 0.00(0.00) | 0.00(0.00) | 0.00(0.00) | 0.00(0.00) |
| | | $\hat{d}$ | 2.00(0.00) | 2.08(0.08) | 2.00(0.00) | 2.08(0.08) | 2.00(0.00) | 2.00(0.00) |
| | 2 | FP | 36.37(19.08) | 15.60(10.75) | 0.00(0.00) | 0.00(0.00) | 0.00(0.00) | 0.00(0.00) |
| | | FN | 0.00(0.00) | 0.00(0.00) | 0.00(0.00) | 0.00(0.00) | 0.00(0.00) | 0.00(0.00) |
| | | $\hat{d}$ | 2.00(0.00) | 2.00(0.00) | 2.00(0.00) | 2.00(0.00) | 2.00(0.00) | 2.00(0.00) |
| | 3 | FP | 25.77(15.83) | 14.49(8.96) | 0.00(0.00) | 0.00(0.00) | 0.00(0.00) | 0.00(0.00) |
| | | FN | 0.00(0.00) | 0.00(0.00) | 0.00(0.00) | 0.00(0.00) | 0.00(0.00) | 0.00(0.00) |
| | | $\hat{d}$ | 1.94(0.24) | 1.97(0.17) | 1.98(0.14) | 1.97(0.17) | 2.00(0.00) | 2.00(0.00) |

In the real data analysis, some selected variables are Median Age (MedianAge), Life Expectancy at Birth (LEx) for both sexes and for males only (LExMale), Life Expectancy at Age 15 (LE15) for both sexes and for females only (LE15Female), Female Life Expectancy at Age 65 (LE65Female), and Male mortality before Age 60 (Q0060Male). Table 2 shows that Group-LASSO selects seven variables, including Median Age (MedianAge), Life Expectancy at Birth (LEx) for both sexes and for males only (LExMale), Life Expectancy at Age 15 (LE15) for both sexes and for females only (LE15Female), Female Life Expectancy at Age 65 (LE65Female), and Male mortality before Age 60 (Q0060Male). While LLA_G and $\text{DASTA}_g$ select a smaller set of variables, including LExMale and LE65Female.

Table 2: First direction $\widehat{\boldsymbol{\beta}}$ captured by Group-LASSO, LLA_G, and $\text{DASTA}_g$, respectively.

| Methods | MedianAge | LEx | LExMale | LE15 | LE15Female | LE65Female | Q0060Male |
|---|---|---|---|---|---|---|---|
| Group-LASSO | 0.0065 | 0.0082 | 0.0617 | 0.0368 | 0.0001 | 0.0714 | -0.0004 |
| LLA_G | 0 | 0 | 0.0790 | 0.0304 | 0 | 0.0794 | 0 |
| $\text{DASTA}_g$ | 0 | 0 | 0.0915 | 0 | 0 | 0.0982 | 0 |

## A.2. Sparse WIRE with Lasso and group-Lasso penalties

Consider an $\ell_1$ penalty on $\boldsymbol{\beta}$, that is $g(\boldsymbol{\beta}) = \|\boldsymbol{\beta}\|_1 = \sum_{i=1}^{d} \lambda_i \|\beta_i\|_1$. This penalty is equivalent to performing multiple Lasso regressions, applying the Lasso penalty to each column of the pseudo response $\mathbb{Y}$. Thus, we solve the following optimization problems for $1 \leq i \leq d$:

$$\hat{\beta}_i = \underset{\beta_i \in \mathbb{R}^p}{\arg\min} \frac{1}{2n} \|\mathbb{Y}_i - \mathbb{X}\beta_i\|_2^2 + \lambda_i \|\beta_i\|_1, \tag{7}$$

where $\mathbb{Y}_i$ is the $i$th column of $\mathbb{Y}$ and $\lambda_i$ is a tuning parameter. The final estimator $\hat{\boldsymbol{\beta}}$ is a matrix composed of $(\hat{\beta}_1, \hat{\beta}_2, \ldots, \hat{\beta}_d)$, and it is referred to as a LASSO WIRE estimator. This approach recovers the central subspace $\mathcal{S}_{Y|X}$ by applying the Lasso penalty to each univariate regression model. The corresponding algorithm is the following.

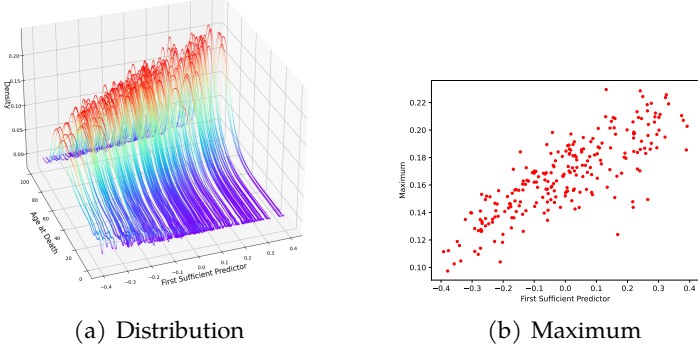

| (a) Distribution | (b) Maximum |
|---|---|

Figure 3: (a) 3D Plot of mortality densities versus the first sufficient predictor; (b) Scatter plot of the maximum value of densities versus the first sufficient predictor.

---

**Algorithm 4:** Algorithm for the LASSO WIRE estimator.

1. Let $\hat{\boldsymbol{\eta}}$ and $\hat{\phi}_1, \ldots, \hat{\phi}_d$ be the first $d$ eigenvectors and eigenvalues of $\hat{\Lambda}$.

2. Let $\mathbb{Y} = -\frac{1}{n-1} D\mathbb{X}\hat{\boldsymbol{\eta}}\mathrm{diag}\{1/\hat{\phi}_1, \ldots, 1/\hat{\phi}_d\}$ and solve the Lasso regression problems,

$$\hat{\beta}_i = \underset{\beta_i \in \mathbb{R}^p}{\arg\min} \frac{1}{2n}\|\mathbb{Y}_i - \mathbb{X}\beta_i\|_2^2 + \lambda_i\|\beta_i\|_1, \quad \text{for } 1 \leq i \leq d.$$

3. $\hat{\boldsymbol{\beta}} = (\hat{\beta}_1, \hat{\beta}_2, \ldots, \hat{\beta}_d)$ is the estimate.

---

Recall that the index set $S$ represents the corresponding active predictors. If the $i$th row of $\boldsymbol{\beta}$ consists entirely of zeros, then the corresponding predictor is inactive, i.e., $i \notin S$. To encourage row sparsity on $\boldsymbol{\beta}$, we consider implementing an alternative penalty to (4), that is group Lasso penalty: $g(\boldsymbol{\beta}) = \sum_{i=1}^{p} \lambda\|e_i^\top\boldsymbol{\beta}\|_2$, where $e_i$ is the $i$th canonical basis vector in $\mathbb{R}^p$. Thus, we consider the optimization problem (8) as follows:

$$\hat{\boldsymbol{\beta}} = \underset{\boldsymbol{\beta} \in \mathbb{R}^{p \times d}}{\arg\min} f(\boldsymbol{\beta}) + \sum_{i=1}^{p} \lambda\|e_i^\top\boldsymbol{\beta}\|_2. \tag{8}$$

Denote the minimizer of (8) as Group-LASSO WIRE estimator. The algorithm is summarized below.

---

**Algorithm 5:** Algorithm for Group-LASSO WIRE estimator.

1. Let $\hat{\boldsymbol{\eta}}$ and $\hat{\phi}_1, \ldots, \hat{\phi}_d$ be the first $d$ eigenvectors and eigenvalues of $\hat{\Lambda}$.

2. Let $\mathbb{Y} = -\frac{1}{n-1} D\mathbb{X}\hat{\boldsymbol{\eta}}\mathrm{diag}\{1/\hat{\phi}_1, \ldots, 1/\hat{\phi}_d\}$ and solve the multitask group-Lasso regression problem,

$$\hat{\boldsymbol{\beta}} = \underset{\boldsymbol{\beta} \in \mathbb{R}^{p \times d}}{\arg\min} f(\boldsymbol{\beta}) + \sum_{i=1}^{p} \lambda\|e_i^\top\boldsymbol{\beta}\|_2.$$

---

To implement the optimization problems presented in Algorithms 4 and 5, we utilize widely-used software packages, namely *glmnet* in R or *scikit-learn* in Python. These packages employ cross-validation to select the optimal tuning parameter $\lambda$ and generate sparse estimates for the central subspace. Hence, we use these methods as benchmarks for comparison purposes. In the subsequent section, we introduce our novel algorithms, which solve optimization problems with non-convex penalties and demonstrate superior numerical performance compared to the benchmark algorithms.

## A.3. Proofs

*Proof of Lemma 1:* Consider the case when $t > t_0$ and want to proof

$$\phi(t) - \rho^*(t) = \phi(t) - \rho(t_0) - (t - t_0)\rho'(t_0) \geq 0$$

To do that, use the convexity of $\phi(t)$, for $t_0 < t_k < t, k \geq 1$, we have

$$\frac{\phi(t) - \phi(t_0)}{t - t_0} \geq \frac{\phi(t_k) - \phi(t_0)}{t_k - t_0}$$
$$\geq \frac{\rho(t_k) - \rho(t_0)}{t_k - t_0}$$

The second inequality dues to that $\phi$ is one maximization function. Let $k$ go to infinity, then

$$\frac{\phi(t) - \phi(t_0)}{t - t_0} \geq \rho'(t_0),$$

which is what we want. Similarly, we can proof $\phi(t) \geq \rho^*(t)$ for $t < t_0$. $\qquad\square$

*Proof of Theorem 1:* Recall that $G(\boldsymbol{\beta}) = f(\boldsymbol{\beta}) + \sum_{i=1}^p \rho(\|b_i\|_2)$, and $N(\boldsymbol{\beta}, \boldsymbol{\beta}^{(k)}) = f(\boldsymbol{\beta}) + \sum_{i=1}^p \rho^*(\|e_i^\top \boldsymbol{\beta}\|_2 \mid \|e_i^\top \boldsymbol{\beta}^{(k)}\|_2)$. Based on Lemma 1, $\rho^*$ is the best maximization function of $\rho$, hence, $\rho(\|e_i^\top \boldsymbol{\beta}\|_2) \leq \rho^*(\|e_i^\top \boldsymbol{\beta}\|_2 \mid \|e_i^\top \boldsymbol{\beta}^{(k)}\|_2)$ and $\rho(\|e_i^\top \boldsymbol{\beta}^{(k)}\|_2) = \rho^*(\|e_i^\top \boldsymbol{\beta}^{(k)}\|_2 \mid \|e_i^\top \boldsymbol{\beta}^{(k)}\|_2)$ for $i \in \{1, \ldots, p\}$. Then we have

$$G(\boldsymbol{\beta}) \leq N(\boldsymbol{\beta}, \boldsymbol{\beta}^{(k)}) \quad \text{and} \quad G(\boldsymbol{\beta}^{(k)}) = N(\boldsymbol{\beta}^{(k)}, \boldsymbol{\beta}^{(k)}).$$

Note that $\boldsymbol{\beta}^{(k+1)}$ is minimizer of $N(\boldsymbol{\beta}, \boldsymbol{\beta}^{(k)})$, hence $N(\boldsymbol{\beta}^{(k+1)}, \boldsymbol{\beta}^{(k)}) \leq N(\boldsymbol{\beta}, \boldsymbol{\beta}^{(k)})$ for any $\boldsymbol{\beta}$.

$$G(\boldsymbol{\beta}^{(k+1)}) \leq N(\boldsymbol{\beta}^{(k+1)}, \boldsymbol{\beta}^{(k)})$$
$$\leq N(\boldsymbol{\beta}^{(k)}, \boldsymbol{\beta}^{(k)})$$
$$= G(\boldsymbol{\beta}^{(k)}).$$

$\qquad\square$

*Proof of Theorem 2:* This is similar to the proof of Theorem 1. Note that $M(\boldsymbol{\beta}, \boldsymbol{\beta}^{(k-1)}) = f^*(\boldsymbol{\beta}, \boldsymbol{\beta}^{(k-1)}) + \sum_{i=1}^p \rho^*(\|e_i^\top \boldsymbol{\beta}\|_2 \mid \|e_i^\top \boldsymbol{\beta}^{(k-1)}\|_2)$. Both $f^*$ and $\rho^*$ are maximization functions of $f$ and $\rho$, respectively. Hence, $G(\boldsymbol{\beta}) \leq M(\boldsymbol{\beta}, \boldsymbol{\beta}^{(k-1)})$ and $G(\boldsymbol{\beta}^{(k-1)}) \leq M(\boldsymbol{\beta}^{(k-1)}, \boldsymbol{\beta}^{(k-1)})$. Also we have $\boldsymbol{\beta}^{(k)}$ is minimizer of $M(\boldsymbol{\beta}, \boldsymbol{\beta}^{(k-1)})$.

$$G(\boldsymbol{\beta}^{(k)}) \leq M(\boldsymbol{\beta}^{(k)}, \boldsymbol{\beta}^{(k-1)})$$
$$\leq M(\boldsymbol{\beta}^{(k-1)}, \boldsymbol{\beta}^{(k-1)})$$
$$= G(\boldsymbol{\beta}^{(k-1)}).$$

$\qquad\square$

