# OpenReview forum: "Sparse Fréchet sufficient dimension reduction via nonconvex optimization"
_CPAL.cc/2024/Conference — CPAL 2024 (Proceedings Track) Oral_

### Official Review · Reviewer_jzrn · 2023-10-07
**Not qualified to review this**

**Rating:** 6
**Confidence:** 1

**Review:**

Thank you for considering me as a reviewer for this manuscript. After careful consideration, I feel that I may not be the most suitable reviewer for this particular work. My expertise does not sufficiently cover the areas related to the theoretical perspective of sufficient dimension reduction, which seems to be a foundational concept in the paper. The work is also defined build on several other Fréchet SDR work that I have never heard of. As such, I am concerned that I might not be able to fully grasp the motivation and nuances of the method presented. It would be in the best interest of the author and AC to not take into account of my comments on this work.

---

### Official Review · Reviewer_LJBD · 2023-10-09
**The theory is interesting. The problem setup is poor.**

**Rating:** 5
**Confidence:** 3

**Review:**

Paper summary: This paper focuses on the problem of sufficient dimension reduction (SDR). The model consists of random object response matrix X in a metric space and a predictor vector Y in a Euclidean space; and SDR assumes that they are statistically independent of each other given an unknown linear function of response matrix X, namely, \beta^T X. The number of features in X exceeds the sample size. The goal is to identify a low-dimensional and sparse representation of the dimension reduction subspace (DSR) --- which is related to \beta. They construct a multitask regression model with synthetic responses (computed from X and Y) and achieve sparse estimation by leveraging the minimax concave penalty. Their algorithms avoid inverting the ill-conditioned covariance matrix. The optimization problems are nonconvex. So they develop and implement a double approximation shrinkage-thresholding algorithm that combines a linear approximation to the penalty term and a quadratic approximation to the loss function. The show the utility of their algorithms on manifold-valued synthetic data and one real-world dataset.


My Evaluation: The problem statement is interesting. However, the authors should reorganize subsection 2.1 to more clearly explain it; perhaps they can dedicate a latex environment under "Problem 1" and formalize the problem. What are the measurements?, what is the goal? and what are the assumptions? One of the main ideas to arrive at the current solution is "conditional mean independence" which is not explained.  Furthermore, it is not clear how one arrives at equation 4 from equation (3). I left a comment below about this. Every step of the derivation should be clearly explained (as well as the required assumptions). Overall, I think this is a nice problem statement. The proposed solutions are neat and, for the most part, clearly explained (section 3). The experiments are limited but overall they showcase the performance of their approach compared to the LASSO-based methods on synthetic data (using small sample sizes). I'm willing in to increase my score if the authors address all the comments in this review.


Questions, Comments, and Suggestions:

(1) In the introduction, the problem of "Fréchet SDR" is not motivated nor introduced.

(2) Equation (1) : What is the orthogonality with respect to? Is there a probability distribution assumption? If so, please make it clear. That way, you can better discuss terms like “regression information”.

(3) (page 1, line 38) “The matrix \beta satisfying (1) is not identifiable.” Up to what? I assume it is upto a post-multiplication by an orthogonal (or an invertible?) matrix . Please make it clear.

(4) (page 1, line 39) “However, DRS is not unique.“ Is this generally true? If so, please dedicate a remark and explain why this the case.  My understanding is that it should depend on the specific model, no? For example, a simple linear model Y = \beta^{T} X + n where n is an independent AWGN. Is DRS unique in this example?

(5)  (page 2, line 81) “The double approximation technique relies less on the initial values and provides explicit expressions at each iteration,  … “ What does it mean that this technique relies less on the initial values? Do you mean it has a global solution? Or maybe it is not sensitive to the initial values? If so, why or in which cases?

(6) (page 3, line 91) "To detect the conditional mean independence, ... " Could you please explain what this means? This seems very important because it is the main reason for using WIRE kernel matrix to identify DSR.

(7) (Proposition 1) . (a) What is "metric space of negative type"? (b) This proposition puts a constraint on both the distribution of X and the underlying metric space. Am I correct? If so, could you please give a couple of examples in which the linearity assumption is true. (c) Please reference the proposition inside its environment.

(8) (Section 2.2) The derived conclusion of \eta \prop 1/n X^T X \beta is true but not immediately obvious. I spent some time deriving it for myself. I suggest that you do the following: (a) Dedicate a definition for the synthetic response variables -- computed from Y and X   (b) Start with the fact that \Sigma Span( \beta ) = Span( \Lambda ) (under the assumptions) and claim that \mathbb{Y} (defined earlier) can be written as \mathbb{Y} = X \beta where \beta is a vector in DSR. Then this can be generalized for when \beta is a matrix. Then, motivate the use of equation 4. The current derivation/explanation is very confusing.

(9) (Line 2 in Algorithm 1). Could you please explain how X^*_j = \lambda / w_j \mathbb{X}_j is derived? This is not obvious to me. A remark on the main idea in adapting LLA to the sparse setting would benefit the readers.  Also, please make sure that your clearly define terms like \mathbb{X}_j, X_j, w_j, e_j, \beta_i, b_{i, k-1},  .... These notations are very confusing. I suggest for rows and columns use e_i^T X , X e_j ( not X_j, ... ) and for iterations use superscript with parentheses like X^{(k)}. Also indices should varying from lower case letters to uppercase ones, like \sum_{n = 1 }^{N}  not \sum_{j=1}^{p} ...

(10) (Algorithm 2) What is O_{i,k+1}?

(11) (Page 6, line 210)  "[24] discovered a disparity a ... " (a) Please do not use numbered references as nouns (everywhere it applies). (b) Please explain why eigenvalues of \Lambda (instead of \Simga^{-1} \Lambda) are important? And also, why do we need the adjustment terms?

(12) The one-step LLA_G algorithm seem to provide very good numerical results (Table 2 and example 3 in Figures 1 and 2 ) compared to DASTA methods. Any comments on the computation complexity of LLA vs DASTA?

(13) (page 1, line 62) "Most of the existing sparse SDR methods employed Lasso or group-Lasso penalty, both of which are convex and lead to biased estimation." What is the source of this biased estimation? And how does the nonconvex MCP penalty remedy that?

---

### Official Review · Reviewer_PKWv · 2023-10-12
**Main Review**

**Rating:** 7
**Confidence:** 2

**Review:**

This manuscript proposes a new method for high-dimensional Frechet SDR by augmenting a weighted inverse regression ensemble with a nonconvex penalty. Overall the reviewer finds this paper easy to follow, and the contributions are clear, hence would like to recommend a “clear accept”. Several minor comments:
1. In line 61, perhaps the authors can consider using a single \cite{} to contain all bibs, so that the citations will appear as [16-27].
2. The reviewer could be wrong, but the formulation in equation (4) looks very similar to a sparse dictionary learning problem. More precisely, for each input data vector $(x_i,y_i)$, we want to use at most $d$ elements $x_i$ to represent $y_i$. The reviewer thinks it would be great if the authors could elaborate more on the connection between the proposed objective function with the sparse dictionary learning problem (especially in the non-convex optimization setting). Note that there is a vast number of papers in the sparse dictionary literature if the authors would like to add some related papers, perhaps the authors could use [1] as a potential starting point. But feel free to ignore this suggestion, if the authors find this suggestion not relevant.

[1] Zhang, Yuqian, Qing Qu, and John Wright. "From symmetry to geometry: Tractable nonconvex problems." arXiv preprint arXiv:2007.06753 (2020).

---

### Meta-Review · Area_Chair_zUhj · 2023-11-11

**Recommendation:** Accept (Poster)
**Confidence:** 4

**Metareview:**

In this paper, the authors deal with the problem of Sparse Fréchet dimension reduction in the setting where features exceed sample size. They propose a nonconvex optimization algorithm that involves a minima concave penalty. The reviewers recognized that the problem under study is interesting and the contributions of the current paper meaningful. The reviewers raised some concerns regarding the organization of the paper and requested clarifications. Moreover, they found somewhat limited empirical results but at the same time admitted that they sufficiently showcase the merits of the approach. The authors responded to all reviewers' comments point-by-point, and their responses were quite convincing. Therefore, I recommend the acceptance of the paper as a poster and strongly encourage the authors to accordingly revise their final submission as requested by the reviewers.

---

### Decision · Program_Chairs · 2023-11-19

**Decision:**

Accept (Oral)

**Comment:**

After a thorough review of the paper titled "Sparse Fréchet Dimension Reduction with Minima Concave Penalty," it is clear that the authors have addressed an interesting problem with meaningful contributions. The paper presents a nonconvex optimization algorithm involving a minima concave penalty in the context of Sparse Fréchet dimension reduction when dealing with feature dimensionality exceeding sample size.

While the reviewers recognized the significance of the problem and the value of the contributions, they had some concerns regarding the paper's organization and requested clarifications. Additionally, they noted that the empirical results, while somewhat limited, still effectively demonstrated the merits of the proposed approach.

During the rebuttal, the authors responded to all reviewers' comments in a convincing manner. Therefore, we recommend the acceptance of the paper. We strongly encourage the authors to carefully revise their final submission in line with the feedback provided by the reviewers.

The action PC chair for this paper is Qing Qu, who made the decision after carefully reading the paper as well as the comments by all reviewers and AC. The decision is agreed upon by all PC chairs.